# High-frequency rectifiers based on type-II Dirac fermions

Libo Zhang[1,2], Zhiqingzi Chen[1], Kaixuan Zhang[1,2], Lin Wang [1✉], Huang Xu[1], Li Han[1,2], Wanlong Guo [1,3], Yao Yang[1,3], Chia-Nung Kuo[4], Chin Shan Lue[4], Debashis Mondal [5,6], Jun Fuji [5], Ivana Vobornik[5], Barun Ghosh[7], Amit Agarwal[7], Huaizhong Xing[2,8✉], Xiaoshuang Chen[1,3✉], Antonio Politano [9,10✉] & Wei Lu[1,3]

The advent of topological semimetals enables the exploitation of symmetry-protected topological phenomena and quantized transport. Here, we present homogeneous rectifiers, converting high-frequency electromagnetic energy into direct current, based on low-energy Dirac fermions of topological semimetal-NiTe$_2$, with state-of-the-art efficiency already in the first implementation. Explicitly, these devices display room-temperature photosensitivity as high as 251 mA W$^{-1}$ at 0.3 THz in an unbiased mode, with a photocurrent anisotropy ratio of 22, originating from the interplay between the spin-polarized surface and bulk states. Device performances in terms of broadband operation, high dynamic range, as well as their high sensitivity, validate the immense potential and unique advantages associated to the control of nonequilibrium gapless topological states via built-in electric field, electromagnetic polarization and symmetry breaking in topological semimetals. These findings pave the way for the exploitation of topological phase of matter for high-frequency operations in polarization-sensitive sensing, communications and imaging.

[1] State Key Laboratory for Infrared Physics, Shanghai Institute of Technical Physics, Chinese Academy of Sciences, Shanghai, China. [2] Department of Optoelectronic Science and Engineering, Donghua University, Shanghai, China. [3] School of Physical Science and Technology, ShanghaiTech University, Shanghai, China. [4] Department of Physics, National Cheng Kung University, Tainan, Taiwan. [5] Consiglio Nazionale delle Ricerche (CNR)- Istituto Officina dei Materiali (IOM), Laboratorio TASC in Area Science, Trieste, Italy. [6] International Centre for Theoretical Physics (ICTP), Trieste, Italy. [7] Department of Physics, Indian Institute of Technology Kanpur, Kanpur, India. [8] Shanghai Institute of Intelligent Electronics and Systems, Shanghai, China. [9] Department of Physical and Chemical Sciences, University of L'Aquila, L'Aquila, AQ, Italy. [10] CNR-IMM Istituto per la Microelettronica e Microsistemi, Catania, Italy.
✉email: wanglin@mail.sitp.ac.cn; xinghz@dhu.edu.cn; xschen@mail.sitp.ac.cn; antonio.politano@univaq.it

The advent of topological phases of matter had a groundbreaking impact on condensed matter physics concerning both the exploration of fundamental physics and novel pathways for technological innovations[1-3]. Among topological materials, Weyl and Dirac semimetals deserve special attention, as they exhibit massless relativistic quasiparticles arising from linear band crossings at the degenerate band crossings protected by crystalline symmetries[4,5]. Topological semimetals represent an ideal platform to explore the exotic physics of dissipation-less carrier-transport in gapless quantum materials, protected by topology.

In contrast to fundamental particles of high-energy physics, the emergent quasi-particles in crystalline materials are not constrained by the Lorentz invariance. This can result in systems with 'tilted' energy dispersion, leading to the new classification of type-I and type-II Dirac/Weyl semimetals (Fig. 1d). The type-II Dirac/Weyl systems possess unbounded electron and hole pockets at the Fermi surface along with a large density of states at the Dirac/Weyl node. This leads to a modulated effective mass and unique properties, such as singular superconductivity and novel quantum oscillations[6,7]. Materials hosting type-II Dirac/Weyl nodes have been recently discovered in transition-metal dichalcogenides (TMDs), e.g., $WTe_2$, $Mo_{1-x}W_xTe_2$, and group-X Pd- and Pt-based ones, enriching the landscape for exploring novel relativistic physics in realistic settings[8-12].

The limitation on the detectable photon energy, imposed by the bandgap, does not apply to these semimetals, and this endows them with broadband photoresponse down to the far-infrared spectral region. Additionally, the existence of titled Dirac/Weyl cones creates a fertile playground for exploring global properties of relativistic quasiparticles manifested via anomalous thermoelectric effect[13], nonlinear anomalous photocurrents[14], novel undamped gapless plasmons[15], and nonlinear optical effects dominated by the Berry curvature dipole in non-centrosymmetric materials[16]. The divergence of the Berry curvature has been observed as a measurable effect from photo-response only at selective wavelengths through single-particle process in mid-infrared regime. The resulting nonlinear response at microwave, terahertz, and far-infrared frequencies[17] is relatively unknown. In addition, for typical tilted Dirac/Weyl nodes in TMDs the degenerate point usually lies far away from the Fermi level (~1 eV in $PtSe_2$[18], ~1.2 eV in $PtTe_2$[19], ~0.5 eV in $PdTe_2$[20], and ~52 meV in $WTe_2$[6]), making it a big challenge to achieve the contribution of the relativistic quasiparticles at low photon energies.

Additionally, the interest towards energy harvesting in the microwave and terahertz frequency range is rapidly increasing, owing to its relevance for wireless technology and portable devices[21]. A high-frequency rectifier, converting oscillating electromagnetic field to a direct current, is a pivotal constituent for sensor and detector technologies deployed in applications, such as telecommunications, bioassays, remote sensing, and quality control, to name a few[22]. Currently, the frequency of rectification in semiconductor junctions or electrical circuits suffers from considerable drawbacks, such as limited transit time, as well as thermal voltage threshold, yielding decreased responsivity at higher frequencies and high manufacturing costs[23]. This represents the main challenge and technological bottleneck to get electrical diodes and photodiodes able to work in the so-called terahertz gap (0.1 to 10 THz).

Here, we demonstrate high-frequency rectification driven by skew scattering in type-II Dirac semimetal $NiTe_2$-based devices, by cohesively and simultaneously manipulating the electromagnetic-field and built-in electric-field. Our $NiTe_2$-based devices display a remarkably high sensitivity even at frequencies higher than those limited by the transit-time, owing to the contribution of topologically protected surface and bulk bands. Additionally, due to the presence of the type-II Dirac nodes close to the Fermi energy and the topological surface states, $NiTe_2$ features high mobility and broadband fast response in the high frequency region[24].

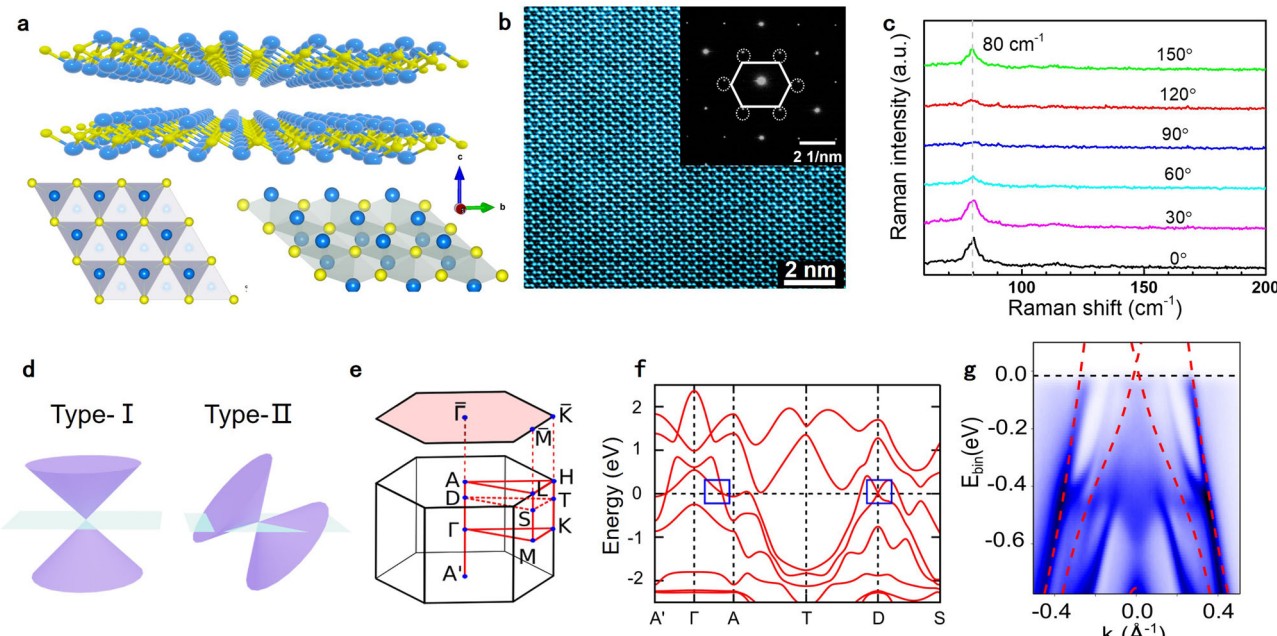

**Fig. 1 Structural and electronic properties of NiTe₂. a** Atomic structure of the $NiTe_2$ transition-metal dichalcogenide, in both side and top views. Yellow and blue balls denote Ni and Te atoms, respectively. **b** Spherical aberration-corrected scanning transmission electron microscopy image. The scale bar is 2 nm. The inset shows the selected area electron diffraction (SAED) pattern. **c** Polarization-resolved Raman spectra collected by exciting the fabricated samples by varying the polarization angle with a 532 nm excitation laser. The observed peak at 80 cm⁻¹ corresponds to the $E_g$ phonon. **d** Schematic of type-I and type-II Dirac fermions. **e** The (001) surface and the bulk Brillouin zone with high symmetry points of $NiTe_2$, and the calculated bulk band structure is shown in **f**. **g** Experimental band structure of $NiTe_2$, measured by ARPES, overlaid with the theoretical bulk bands represented by red-dashed lines.

## Results

**Materials growth and angle-resolved photoemission spectroscopy (ARPES).** $NiTe_2$ crystallizes in a centrosymmetric trigonal crystal structure, depicted in Fig. 1a, belonging to the $p\bar{3}ml$ space group and $D_{3d}$ point group[25]. The superb crystalline quality with negligible number of defects is confirmed by the analysis of different experiments such as high-resolution transmission electron microscopy (HRTEM) (Supplementary Fig. 1a), spherical-aberration corrected scanning transmission electron microscopy (STEM) in (Fig. 1b), and by electron diffraction patterns acquired with low-energy electrons (LEED) (Supplementary Fig. 1d) or from small-area (SAED) (inset of Fig. 1b).

The presence of a pair of type-II Dirac nodes along the $C_3$ rotation axis, with quadruple degenerate band-crossing due to the existence of both inversion and time reversal symmetries, is depicted in the theoretical band structure shown in Fig. 1f, which was validated by angle-resolved photoelectron spectroscopy (ARPES) data (Fig. 1g). Notably, the bulk Dirac point in $NiTe_2$ is in close proximity to the Fermi energy ($E_F$), in contrast to other TMDs, such as $PdTe_2$, $PtSe_2$, and $PtTe_2$[26].

The anisotropy of the atomic structure and the corresponding electronic band structure is reflected in the experimental photoresponse (Supplementary Fig. 2a). To identify the anisotropic crystal orientation, we carried out experiments by polarization-resolved Raman spectroscopy, in which the intensity of the $E_g$ phonon at 80 $cm^{-1}$ varied with the polarization angle (Fig. 1c and Supplementary Fig. 1c).

**Device design and high-frequency rectification characteristic.** $NiTe_2$-based high-frequency rectifiers are devised in the form of planar structure with electrically connected log-periodic antenna fabricated in symmetrical geometry (Fig. 2a). This configuration converts the incident electromagnetic field into localized oscillating electric field via the so-called spoof-plasmon effect[27–29]. The oscillating electric field near the metal-$NiTe_2$ interface is driven by the alternating displacement-current across the channel-gap ($L = 6\ \mu m$ here) in Fig. 2b. The peak of the photocurrent distribution is validated to be near electrodes by near-field THz photocurrent microscopy in recent works, due to the strongest interaction near the material–metal interface[30–33]. Due to the contribution from (i) topological surface states, (ii) localized electric-field, and (iii) strongly asymmetric dispersion near the bulk Dirac point, it is possible to obtain broadband absorption at microwave and THz frequencies[34]. Note that the top surface has a predominant weight in the photocurrent, due to the static screening of the potential at the metal–material interface[35]. The time-resolved photocurrent at different electromagnetic frequencies from 0.04 to 0.30 THz was obtained with modulated ON/OFF sources for devices working in the self-driven mode (see Supplementary Fig. 2c and Methods section). The largest obtained values of responsivity were 12.57, 5.72, and 0.25 A $W^{-1}$ at 0.04, 0.12, and 0.3 THz, respectively (Fig. 2c). The linear input-power dependence of the photocurrent at different bias voltages (Fig. 2d) indicates a large dynamic regime and a second-order nonlinear response in our devices. The low excitation power threshold of 1 nW, which refers to the minimum input power to surmount the potential barrier, highlights the lower noise floor of the entire measurement platform. Concurrently, the devices display a response time of 2.7 μs, i.e., several orders of magnitude faster than other room-temperature thermal-based sensors (~tens of ms) (Supplementary Fig. 2d)[36], due to the more efficient carrier collection under the high-frequency rectification. The definite features of the second-order nonlinear rectification studied here are the self-driven photocurrents at all incident frequencies, obliterating the technological bottlenecks inextricably linked to

high dark current, cryogenic cooling, and high power required in traditional photodetector[37]. With the bias voltage traversing across the channel, the nonequilibrium electrons produced under action of strong localized field generated by an oscillating electromagnetic field are accelerated unilaterally along the channel (see Fig. 2a), leading to a significant improvement in the responsivity.

As another figure of merit, the dark noise related to thermal agitation of charge carriers, as well as the shot noise in a typical planer device, determine the sensitivity of detector. To evaluate the general performance of the detectors, we analyzed the noise spectra of the device as a function of frequency, and calculated the noise-equivalent power (NEP), which is the minimum detectable power for a signal-to-noise ratio of 1 (see Supplementary Fig. 2e for noise current spectral density). As reported in Fig. 2g, the experimentally derived minimum NEP is 4.9 pW $Hz^{-1/2}$ at 0.04 THz, 19.6 pW $Hz^{-1/2}$ at 0.12 THz, and 89.8 pW $Hz^{-1/2}$ at 0.30 THz, respectively, consistent with the theoretical estimation (dashed line in Fig. 2g, see Methods for more information). Remarkably, the device performance remains stable and the devices maintain fast response time even after exposure in ambient conditions for one month. The total current in Fig. 2f and the resistance in Supplementary Fig. 2 showed no noticeable modification with time.

**Nonlinear rectification current in $NiTe_2$.** To understand the linear power dependence of the experimental rectification current on the input excitation power in Fig. 2d, we turn to a microscopic interpretation following the Boltzmann kinetic equation for the carrier distribution function $f_k(\varepsilon)$ under a spatially homogeneous electric field $\mathbf{E}$. In $NiTe_2$, the spin-momentum-locked surface charge carriers break the inversion symmetry. As a consequence, asymmetric transition rates exist between the k-k' states and the -k and -k' states, i.e., $W_{k\ k'} \neq W_{-k,\ -k'}$, where $W_{k\ k'}$ represents the probability of the transition of a carrier with momentum k to momentum k' after scattering. It includes the symmetric ($W^s_{kk'}$) and the asymmetric ($W^a_{kk'}$) transition rates, $W_{k\ k'} = W^a_{kk'} + W^s_{kk'}$. The asymmetric $W^a$ is finite only in systems with inversion symmetry breaking, and it can generate a dominating contribution by means of the skew scattering of chiral Bloch electrons even in the system with spin-rotational symmetry[38], resulting in an effective D.C. rectification current. Here, we introduce the nonlinear photo-response tensor $\sigma^3_{abc}$, where the indices a, b, and c refer to the coordinate axes. The second-order nonlinear response is described by the expression, $j_a = \sigma^3_{abc} E_b^*(\omega) E_c(\omega)$, where $\sigma^3_{abc}$ is the third-rank nonlinear conductivity tensor, which depends on the symmetries of the system and $E$ is the component of the ac electric field at frequency $\omega$. In bulk crystals with $D_{3d}$ point group symmetry, the tensor $\sigma$ has only two independent linear components, which can be expressed by the nonvanishing elements $\sigma_{aac} = \sigma_{bbc}$, $\sigma_{aab} = \sigma_{baa} = -\sigma_{bbb}$[39]. Accordingly, we can easily deduce that the induced rectification current at zero bias depends on the components of the linearly polarized light incident beam via the equation $j = E_x^2 - E_y^2$, where $E_x$ and $E_y$ are the incident electric field components along the coordinate axes aligned with the symmetry axes of the crystal (Details in supplementary note 6). When a finite bias is applied, the nonlinear photo-response is proportional to the applied direct electric field $E_{DC}$, as the effect of the non-equilibrium electrons in the applied bias window is added to the zero-bias response, according to: $j_a = \sigma'^3_{abc} E^*_b(\omega) E_c(\omega) E_{DC}(\omega = 0)$. Our insight is also validated in the low-temperature experiment (Supplementary Fig. 4g), in which the trends of temperature-dependent photocurrent are different between zero-bias and above 10 mV bias modes. That is to say, the total photoresponse can be decomposed as the bias-

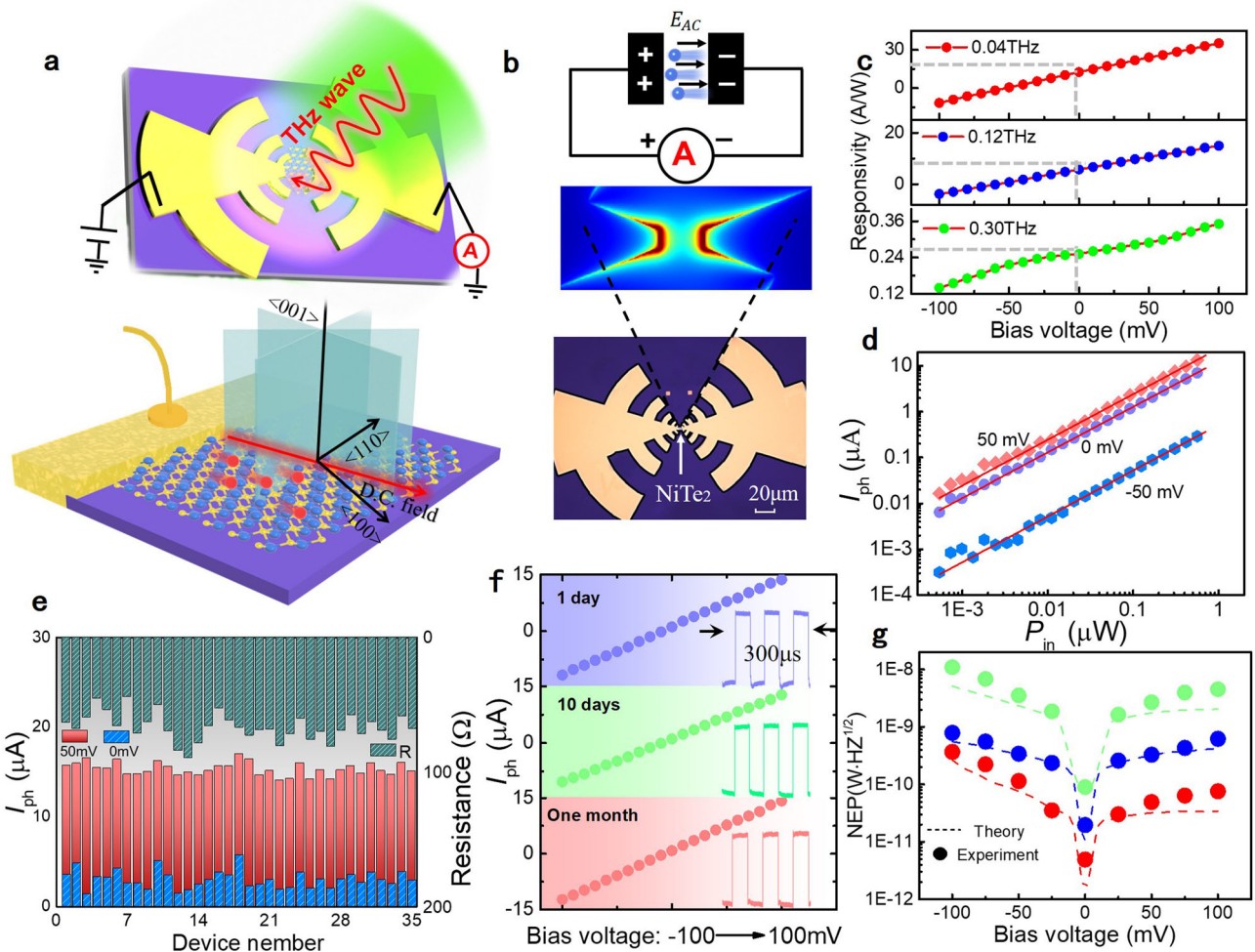

**Fig. 2 Schematics of device structure and high-frequency rectification characteristics. a** Schematic representation of the experimental setup, and the nonequilibrium carrier-diffusion following the DC field-induced symmetry breaking across the channel. **b** Localized field distribution (middle) near the metal–NiTe$_2$ interface, as a result of the ac displacement-current oscillation induced by the high-frequency electromagnetic radiation (upper). The optical micrograph of the device is shown at the bottom. **c** The bias-dependence of the responsivity at different frequencies: 0.04, 0.12, and 0.30 THz. **d** Electromagnetic power dependence of rectified current at different bias voltages, with the lines being the linear fitting. **e** Measured resistance (green bars) and rectified currents (red bars: 50 mV bias, blue bars: zero bias) obtained from dozens of devices, highlighting the excellent reproducibility of our experiment. **f** Device performance recorded after different exposure periods in ambient environment, with the inset showing the corresponding transient time-response. **g** Room-temperature noise-equivalent power (NEP) at different bias voltage. A satisfactory agreement is found between experimental results (dotted lines) with theoretical predictions (dashed lines).

dependent third-order process superimposed on the zero-bias response. The third-order process can be interpreted as that the nonequilibrium carriers under electrical bias are accelerated unilaterally from one side to another side of the channel, due to the bias-induced asymmetry, resulting in a linear growth of photocurrent. In addition, it could also be understood that the electric field $E_{DC}$ plays the main role in tilting the Fermi levels[40], which results in the differently allowed momentum spaces for nonequilibrium carriers generated from opposite Dirac nodes in k-space, so that the non-equilibrium states between these nodes cannot cancel out and will contribute to the net third-order photocurrent, as tunable by $E_{DC}$.

To eliminate possibility of some unknown artifacts causing rectification in our experiments, we fabricated dozens of devices using high-precision electron-beam lithography on the selected area of samples which have excellent crystalline quality as confirmed by atomic force microscopy (Supplementary Fig. 3). However, all devices showed similar characteristics, as highlighted in Fig. 2e. Remarkably, the net photocurrent and resistance were

still preserved after exposing the devices to environmental conditions for a long time, without any significant decrease in the magnitude. Both these facts rule out the possibility that the photocurrent could be caused by random fluctuations in the device.

Having ruled out random artifacts as the source of rectification and second-order optical response in NiTe$_2$, we considered the role of its surface states for the possibility of inversion symmetry breaking. The surface band structure of the as-cleaved (001) surface of NiTe$_2$, measured by ARPES for high-symmetry directions (Fig. 3a), exhibits several surface states along the $\bar{\Gamma} - \bar{M}$ direction of the Brillouin zone, including a Dirac-like conical crossing at a binding energy of $-1.4$ eV. Correspondingly, we show the existence of inverted band gaps (IBGs) formed by the bands of opposite parity (Fig. 3b) in the valence band at the A point, implying the existence of topological surface states. The Dirac node in the surface states appears from the lower IBG (see Supplementary Fig. 5 for more details about the evolution of the Te 5$p$ orbitals).

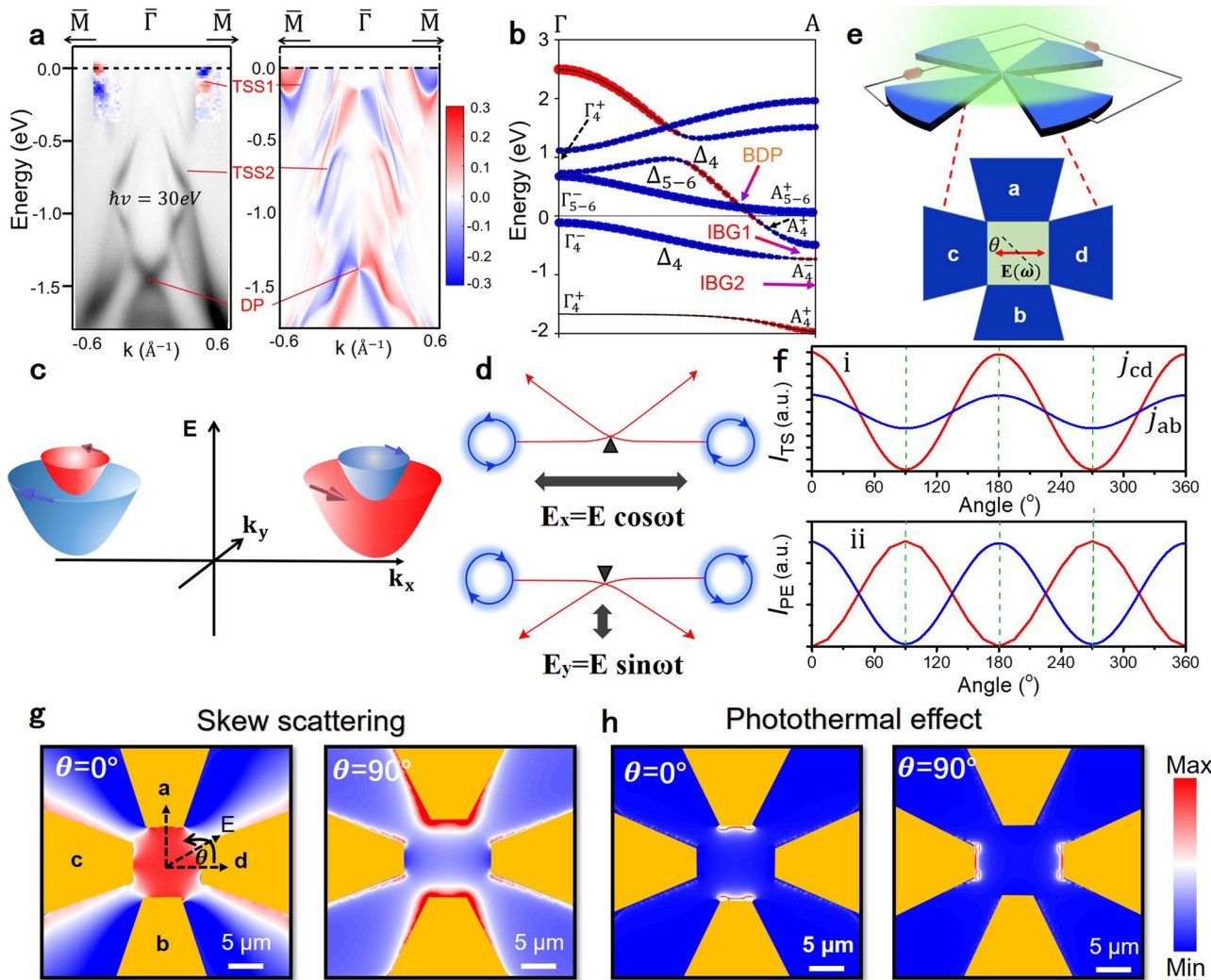

**Fig. 3 Theoretical analysis of second-order rectification current in NiTe₂. a** The experimental ARPES spectrum (left panel) along the $\bar{\Gamma} - \bar{M}$ direction with red and blue areas depicting the oppositely spin-polarized states, along with the corresponding calculations for the spin-polarized surface spectral function (right panel). Theoretical predictions capture the observed topological surface states (TSS), along with the measured spin texture reasonably well. **b** Orbital-resolved band structure and different band inversions along the $\Gamma$ - A direction, along with the irreducible representations of the involved electronic states. The bulk Dirac point (BDP) and a pair of inverted band-gaps (IBG1 and IBG2) can be clearly seen. **c** Schematic of the spin-polarized TSSs at the Fermi energy in NiTe₂ which also show helical spin-momentum locking. **d** Trigonal crystal-field scattering of chiral surface Bloch-electrons driven by different ac electric field components, generating a net photocurrent. **e** Schematics of a rectifier based on four-terminal sector antenna under electromagnetic radiation with the polarization angle θ. **f** Derivatives of rectified photocurrent at different incident polarization, generated (i) by the skew scattering of chiral Bloch electrons and (ii) by the photothermal effect generated by the non-equilibrium distribution of the carriers. Comparisons for theoretical results of local distributions of rectified current at specific polarization angle θ = 0° and θ = 90° are shown in **g** for trigonal skew-scattering and in **h** for photothermal effect. The gradual color contour reflects the localized electric-field enhancement strength distribution. The THz output source direction in which the light polarization direction is parallel to the c–d direction is defined as 0°, and the light polarization direction moves counterclockwise. The scale bar is 5 μm.

Considering the topological origin of these surface states, we used the spin-polarized ARPES measurements to explore the chiral spin-texture, and the crossover of the two-opposite spin-polarization of almost equal magnitude for the surface states can be clearly seen. More importantly, there are also spin-polarized surface states near the Fermi energy (Fig. 3a, c), which originate from a topological band inversion in the conduction band and display helical spin-momentum locking. The inversion-symmetry broken surface states in NiTe₂ can combine with the chiral Bloch electrons to induce skew scattering under trigonal crystal-field (a strong trigonal crystal field generated by the layered crystal structure of NiTe₂, which separates $p_z$ orbitals from the $p_x$, $p_y$ orbitals. See Supplementary note 3 for details), which produces

second-order rectification current. Specifically, when the device is irradiated by electromagnetic radiation, the chiral Bloch electrons are driven back and forth by the ac oscillation of an electric field near the scattering site of trigonal crystal-field. Due to the spin-momentum locked TSSs, the scattered wave-packet will shift towards the same direction even with different spin-rotational directions, leading to the alignment of excess flow (Fig. 3d). In this regard, this phenomenon can be observed when the incident frequency is lower than the skew-scattering rate (or they are close), so that the stationary directional photocurrent can be formed in the process after momentum relaxation. The electro-magnetic frequency studied here could meet the requirement of the timescale to establish the rectification during the skew-

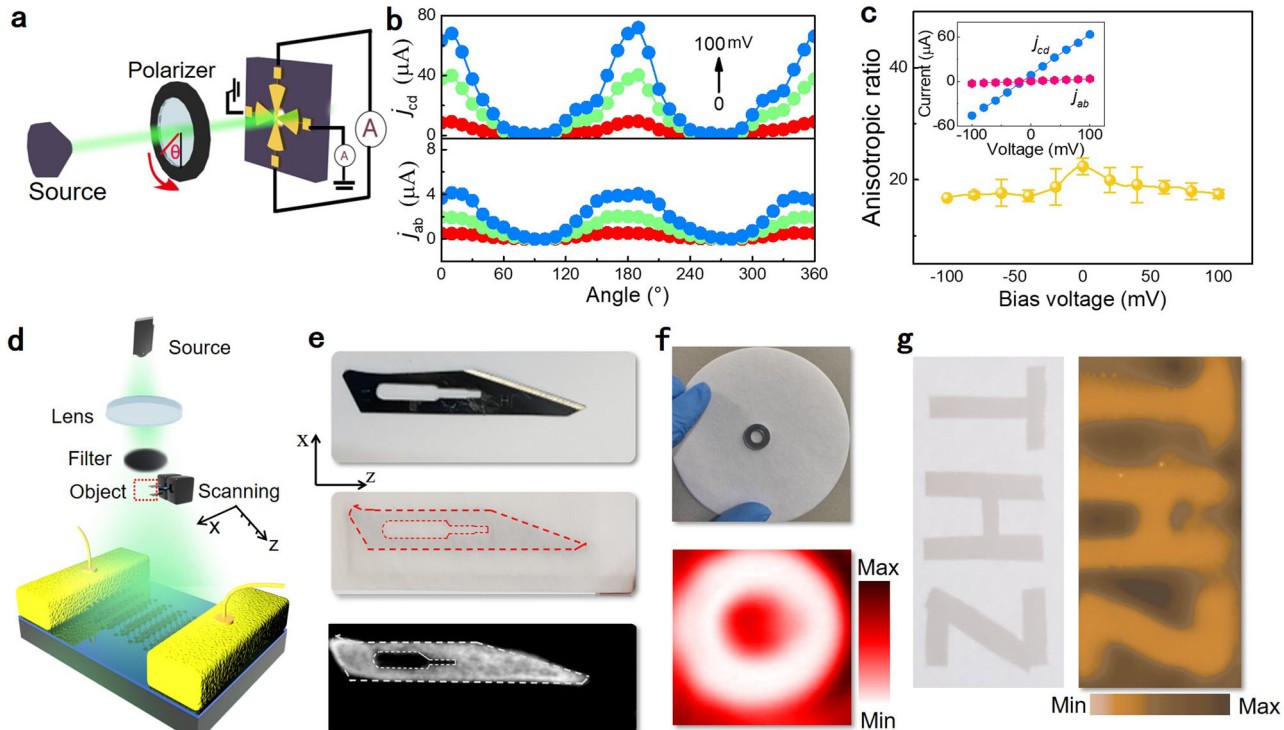

**Fig. 4 High-frequency anisotropy photocurrent and imaging application. a** Schematic illustration of the polarization-sensitive photocurrent measurement. **b** Polarization-angle dependence of the photocurrent response along the x, y direction at different bias voltages. The angle of the polarizer rotates clockwise with respect to the polarization orientation of the incident light. The anisotropic ratio is defined as the ratio of the photocurrent response in the y direction to that in the x direction. **c** Different photocurrent corresponding to two mutual vertical electrodes increase with the bias voltage. Anisotropic photocurrent response of device along two orthogonal crystalline directions with bias added up to ±100 mV. **d** Optical path diagram based on THz imaging. **e–g** Photographs of the enclosed blade, copper lettering "THz" and metallic ring, and their raster scanning imaging at 0.3 THz. The objects were clearly revealed in an envelope, which was invisible to the naked eye. The NiTe$_2$-based detector was biased at 10 mV during imaging experiments.

scattering[41] and the photoresponse will decline at higher frequency.

As a consequence, the skew scattering of the chiral inversion symmetry broken surface states in NiTe$_2$ produces a net photocurrent proportional to $E_x^2 - E_y^2$ (Supplementary note 6) which is consistent with our experiment and above symmetry analysis (shown in Supplementary Fig. 2a). This is inherently different from the local electron-temperature driven photocurrent initiated by accidental symmetry breaking following only the electric field intensity distribution $|E|^2$ [42]. To explore this further, we calculated the particular signatures of high-frequency rectification, due to the skew scattering of the chiral surface wave-packets by following the distribution of oscillating electromagnetic field in crossly arranged electrodes (Fig. 3e). The obtained results are shown in Fig. 3f–i, which highlight that there is no substantial phase-shift between net photocurrents along the a–b ($j_{ab}$) and c–d ($j_{cd}$) directions, whenever the skew scattering of the surface chiral Bloch electrons under trigonal crystal field dominates. In contrast, the polarization-dependent photothermal effect caused by the accidental symmetry-breaking, generates a π/2 phase-shift in Fig. 3f–ii. Furthermore, the rectified photocurrent at zero-bias decreased by a factor of two for the skew scattering mechanism at lower temperature, while the photothermal effect increases on decreasing temperature (Supplementary Fig. 6c)[43]. Specifically, the redistributed electric field due to the presence of the metal electrodes can be simulated to distinguish the two effects. Remarkably, the photocurrent distribution of the photothermal effect has a π/2 phase-shift at any radiation polarization angle (0° and 90° in Fig. 3h). In contrast, the photocurrent dominated by skew scattering of chiral

Bloch electrons under trigonal crystal-field has no phase-shift (Fig. 3g). Our experimental data for the rectification current show the following features: (i) $j = E_x^2 - E_y^2$ or $j = \cos 2\theta$ (Supplementary Fig. 2a), (ii) no phase lag between the crossed currents $j_{ab}$ and $j_{cd}$ (see Fig. 3f (i) for simulation and Fig. 4b for experiment), and (iii) the zero-bias rectification current decreases with decreasing temperature (Supplementary Fig. 5b). These features rule out the possibility of a predominantly photo-thermal current, indicating the skew scattering of the surface chiral Bloch electrons as the most probable origin of the second-order rectification current.

Finite side-jump process might affect the second-order rectification current through modifying the second-order conductivity, due to the interplay of Berry curvature and scattering[44]. However, as NiTe$_2$ is a Dirac semimetal, it does not have a finite Berry curvature in the bulk. The skew scattering contribution is still dominant contribution (Details see supplementary note 7).

**Rectified high-frequency photocurrent with anisotropy.** Now, we turn to characterize the anisotropic properties of the high-frequency rectifier by following the scheme in Fig. 4a. To this aim, we guided the electromagnetic radiation normally to the device after the polarization rotator. The beam size was much larger than the device to ensure the uniformity of the power intensity on the device. Linearly polarized light with electric field oriented along different angles θ from 0° to 360° was used to explore the intrinsic behavior of the anisotropic rectification current in NiTe$_2$. Notably, the rectified signal, originating from the interaction between the polarized incident field and anisotropic trigonal-scattering of the surface states, exhibited a large anisotropy ratio $j_{cd}$ / $j_{ab}$ > 22

(Fig. 4c, see Supplementary Fig. 7 for the resistances along the a–b and c–d directions and the optical micrograph), congruently with the anisotropic nature of the electronic states in NiTe$_2$. The photocurrents recorded from the two pairs of electrodes (cd and ab in Fig. 4c.) increased by nearly an order of magnitude for an applied bias voltage of 100 mV (see Fig. 4b).

When applying a bias voltage, the anisotropy ratio decreased from 22 to 18. This deviation is inextricably related to the bias-induced photoconduction shown in Fig. 2c. A similar anisotropic behavior at infrared wavelength was observed in non-centrosymmetric type-II Weyl semimetals $T_d$-MoTe$_2$ and $T_d$-WTe$_2$. We note that this feature is useful for potential applications connected with nonlinear phenomena, second harmonic generation and polarimetry imaging[45]. However, there is an innate difference between Weyl-like and Type-II Dirac semimetals. Explicitly, in Weyl semimetals the rectification current is caused by the singularities of the Berry curvature and it is accompanied by low excitation power whereas in Dirac semimetals the Berry curvature is absent[17]. The rectification current in Dirac semimetals arises from the scattering of the surface states.

### High-frequency imaging application of NiTe$_2$-based device.
To explore the application capability of our NiTe$_2$-based rectifier for high-frequency electromagnetic imaging, we placed a metallic ring and a knife blade inside an envelope as test objects. We focused an electromagnetic wave at 0.3 THz by means of two pairs of off-axis parabolic mirrors, and the images were acquired by raster-scanning the object at the beam focus, consisting of $100 \times 100$ points with 20 ms integration time at every point (Fig. 4d). Even though the objects were enclosed and were invisible to the naked eye, their presence is evident in our imaging experiments (Fig. 4e–g). These results demonstrate that our device is already exploitable for large-area and fast imaging of macroscopic objects in a realistic setting.

### Discussion
Transition-metal dichalcogenide NiTe$_2$ hosts type-II Dirac fermions in vicinity of the Fermi energy and, accordingly, it represents an interesting platform to explore exotic carrier response of topological semimetals at low photo-excitation energies. We have explicitly shown that the polarization-angle-dependent second-order non-linear photoresponse in NiTe$_2$ arises from the asymmetric scattering in the inversion-symmetry broken surface states of NiTe$_2$. This has practical application for fast response capability and highly sensitive detection, in the field of room-temperature THz technology. The obtained responsivity of 0.25 A W$^{-1}$ of the rectification effect in the self-powered mode is far larger than that observed in other similar materials even in a simple configuration. The observed photoresponse can be tuned by means of doping the bulk crystal to change its bulk and surface properties. Our results combine the physics of topological semimetals and non-linear optical phenomena to open up the possibilities for exploring new fundamental physics and high sensitivity applications in the field of THz photonics and optoelectronics.

### Methods
**Single-crystal growth**. Single crystals of NiTe$_2$ were grown by the Te flux method. Mixtures of high-purity Ni powder (99.99%) and Te ingots (99.9999%) were vacuum sealed in the quartz tube. The quartz ampoule was heated to 1050 °C for 10 h, dwelled for 10 h, then slowly cooled to 600 °C with a rate of 3 °C/h and annealed at 600 °C for 100 h to improve the quality of crystals. The remaining Te flux was removed by centrifuging above 550 °C and finally several shiny plate-like single crystals with a typical size of $8 \times 8 \times 1$ mm$^3$ were obtained, with the flat surface of the crystal corresponding to the (001) plane.

**Characterization of grown crystals**. The absence of contamination is confirmed by the analysis of experiments by energy-dispersive X-ray spectroscopy (EDS) and element mapping, as well as by the featureless vibrational spectrum (Supplementary Fig. 1b). The low-energy electron diffraction (LEED) (Supplementary Fig. 1d) pattern reveals well-resolved spots with hexagonal symmetry, revealing the excellent crystalline order of the as-cleaved surface.

**Density functional theory**. We performed all the density functional theory (DFT)-based ab initio electronic structure calculations with the projector-augmented-wave (PAW) pseudopotentials and a plane wave basis set using the VASP package. The exchange correlation part of the potential was treated within the generalized gradient (GGA) approximation framework developed by Perdew–Burke–Ernzerhof (PBE). We used 500 eV energy cutoff for the planewave basis set. A k-grid of $12 \times 12 \times 8$ was used for the momentum space integration. We relaxed the cell parameters and the atom positions until the residual force on each atom becomes less than 0.001 eV/Å. The relaxed lattice parameters ($a = 3.850$ Å and $c = 5.260$ Å) match very well with the experimental lattice parameters. Surface energy spectrum was obtained within the iterative Green's function method implemented in the Wannier Tools package.

**ARPES experiments**. ARPES measurements on the NiTe$_2$ single crystals were performed at the APE-LE beamline of ELETTRA Synchrotron in Trieste, Italy, using DA30 electron energy analyzer. Photon energy was 21 eV. Energy and angular resolution were 15 meV and 0.2°, respectively.

**Device fabrication**. NiTe$_2$ flakes were mechanically exfoliated by means of an adhesive tape from bulk NiTe$_2$ crystal and successively transferred onto a high-resistance silicon wafer covered with an insulating 300 nm SiO$_2$ layer. Standard ultraviolet lithography technique was used to pattern electrodes, followed by electron-beam evaporation of 5 nm Cr / 70 nm Au. For the device with a four-terminal sector antenna, NiTe$_2$ flakes were transferred to the engraved electrode by dry transfer to complete a four-terminal sector antenna configuration.

**Photocurrent measurements**. The current–voltage characteristic curve of the photodetector was measured by using precision Keysight current–voltage analyzer at room temperature. For the photocurrent measurement, THz frequency was tuned up to 0.30 THz (WR 2.8 Tripler) output from am IMPATT 100 GHz Diode, and 0.12 THz (WR 9 Tripler) from VDI multiplier connected to a 40-GHz microwave source. The 0.3 THz output beam was collimated by a set of two Polymethylpentene (TPX) lenses with 100 mm effective focal length (EFL) resulting in a 5 mm diameter focal beam spot while its amplitude was modulated as a square wave at 1 kHz. The power output, calibrated by a TK100 power-meter, was 100 μW. The photoresponse of our device was recorded under closed-circuit configuration by means of lock-in amplifier technique. Data of response time were obtained directly from the high-speed sampling oscilloscope. The detector responsivity $R_A$ was retrieved from $I_{ph}$ via the relation $R_A = I_{ph}/P\,S_a$, where $P$ is the output power intensity, $P_{in}$ is input power density ($P_{in} = P\,S_a$), $S_a$ is the active detection area. The effective detection area of our devices, $S_a = 2.8 \times 10^4$ μm$^2$, was smaller than the diffraction limited area[46] $S_\lambda = \lambda^2/4\pi$ at 0.04, 0.12, and 0.3 THz, so the active detection area was taken as $S_\lambda$ at 0.04, 0.12, and 0.3 THz. The NEP was evaluated as $v_n / R_v$, where $v_n$ is the root mean square of the noise voltage and $R_v$ is the voltage responsivity: $R_v = r \times R_A$, $r$ is device resistance. To provide the lower limit of noise figure, both the thermal noise $v_t$ and shot noise $v_b$ were extracted from the electrical characteristic of device via $v_n = (v_t^2 + v_b^2)^{1/2} = (4k_B Tr + 2q I_d r^2)^{1/2}$, where $k_B$ is Boltzmann constant, $T$ is temperature, q is the elementary charge, and $I_d$ is the bias current or dark current of the device.

### Data availability
All technical details for producing the figures are enclosed in the supplementary information. Data are available from the corresponding authors L.W. or A.P. upon request.

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

## Acknowledgements

The support was provided by the State Key Program for Basic Research of China (Nos. 2017YFA0305500, 2018YFA0306204), the National Natural Science Foundation of China (Nos. 61521005, 61875217, 91850208), and the STCSM Grants (No. 1859078100, 19590780100). A.A. and B.G. acknowledge the Science Education and Research Board (SERB) and the Department of Science and Technology (DST) of the government of India for financial support. A.A. and B.G. acknowledge the High Performance Computing facility at IIT Kanpur, for computational support. Y.Y. acknowledges the support from Analytical Instrumentation Center (#SPST-AIC10112914), Soft Matter Nanofab (# SPST-SMN180827), and Quantum Device Lab, Shanghai Tech University. D.M. acknowledges the receipt of a fellowship from the ICTP Programme for Training and Research in Italian Laboratories, Trieste, Italy. The Trieste team acknowledges that the nanoscience foundry and fine analysis (NFFA) project, i.e., "I.V., J.F., and D.M". The project was funded by State Key Laboratory for Modification of Chemical Fibers and Polymer Materials, Donghua University (KF1809).

## Author contributions

L.B.Z. and L.W. wrote the manuscript, with the support of A.P., A.A., and B.G. for the parts related to physicochemical and electronic properties. H.Z.X., X.S.C., and W.L. supervised the project and discussed with the experimental results. L.B.Z., W.L.G, and Z.Q. Z.C. performed the cryogenic measurement of photoresponse. L.B.Z. fabricated all devices. H.X. completed the STEM characterization. Y.Y. made polarized Raman and AFM characterization. L.H. and K.X.Z. assisted in the theoretical simulation of the antenna. B.G. and A.A. provide first-principles calculations. ARPES experiments were carried out by D.M., J.F., A.P., and I.V. Samples were grown by C.S.L., C.N.K., and A.P. All authors commented and discussed on this work.

## Competing interests

The authors declare no competing interests.
