## [Peer Review File · Nature Communications]

REVIEWER COMMENTS

Reviewer #1 (Remarks to the Author):

I have carefully reviewed "High-frequency rectifiers based on type-II Dirac fermions" by Libo Zhang et al. This MS reported a large THz induced photocurrent in topological semimetal-NiTe₂ at room-temperature with a photosensitivity as high as 251 mA W⁻¹. The performances of the detector exhibit broadband operation, high dynamic range, as well as high sensitivity. The theoretical discussion is sufficient and indeed explain the observed phenomenon. Since the work represent a novel demonstration of THz induced dynamics in Weyl materials, in principle I can recommend its publication in Nature Communications after the following comments have been addressed.

1. The author should state in the main text what is the THz source, its power output, and the size of the focal spot. It is not clear to the reader.
2. The bulk of the supplemental section 7 should be discussed in the main text. It is also not clear how the current anisotropy comes from the calculation.
3. Why the photocurrent is observed at THz frequencies is not clearly explained. It would be better if the authors can add a discussion on the frequency dependence.
4. The sensitivity of the material can probably promise near-field THz photocurrent measurement. THz s-SNOM imaging can directly probe the proposed orientation dependent photocurrent, especially at the vicinity of the electrodes. Maybe the authors can comment on this?
5. Many sentences can benefit from modifications/improvements in language, for example:
Line 91 "Our NiTe₂-based devices display a remarkably high sensitivity even at frequencies higher than those limited by the transit-time, with significant involvement of topologically protected surface and bulk bands." It is not clear what does the authors mean by "significant involvement"
Line 313 "In Weyl semimetals, the rectification current is caused by the singularities of the Berry curvature and it is accompanied with poor power-linearity, whereas in Dirac semimetals there is not Berry curvature." ♦ There is no Berry curvature.

Reviewer #2 (Remarks to the Author):

The manuscript considers second-order response in a topological semimetal NiTe₂ for a high-frequency rectifier. The authors performed materials characterization both experimentally and theoretically and fabricated a rectifier device to show rectification and imaging in the THz frequency range. The use of a strongly-tilted type-II Dirac semimetal provides an interesting direction to this field. Overall, I think that the manuscript contains intriguing results supported by sound analyses. I would recommend for publication with some questions and comments below adequately answered.

1. The authors find the bulk Dirac point near the Fermi energy and state that it makes "a big challenge to signify the contribution of relativistic quasiparticles at low photon energies." As they use the topological surface state for rectification, the importance of the Dirac point close to the Fermi surface is not evident. They could mention the reason why it has to be.
2. In line 154, they wrote, "The definite features of the nonlinear effects leading to the rectification are voltage-independent photocurrent at all incident frequencies". The meaning of this statement is unclear. They mentioned that applying a bias voltage does have some effects via the third-order process.
3. Topological surface states are expected to appear both top and bottom surfaces. Presumably, only one surface contributes, or the contribution of one surface dominates. The authors should comment on the two-surface contributions.
4. They attributed the observed rectification to skew scattering. It may be the most likely mechanism,

but with only their observations, we cannot exclude other possibilities such as side jump due to symmetry reasons. Any comments would be helpful for readers.

5. I could not see the reason why the zero bias rectification current decreases at low temperatures. It seems that thermally-smeared Fermi distribution degrades the effect at high temperatures.

6. The schematic figure in Fig. 3d needs explanation. Also, Fig. 3g,h do not have a figure legend or detailed description, so that the meaning is unclear.

7. In line 314, the authors mentioned the poor power linearity in Weyl semimetals without details or citations. It should be explained.

————— REPLY TO REVIEWER 1 —————

REFEREE: I have carefully reviewed “High-frequency rectifiers based on type-II Dirac fermions” by Libo Zhang et al. This MS reported a large THz induced photocurrent in topological semimetal-NiTe₂ at room-temperature with a photosensitivity as high as 251 mA W⁻¹. The performances of the detector exhibit broadband operation, high dynamic range, as well as high sensitivity. The theoretical discussion is sufficient and indeed explain the observed phenomenon. Since the work represent a novel demonstration of Thz induced dynamics in Weyl materials, in principle I can recommend its publication in Nature Communications after the following comments have been addressed.

AUTHORS’ REPLY: We thank the reviewer for carefully reviewing our manuscript, for appreciating our paper and for his/her very constructive comments. We find the suggestions of the referee to be extremely helpful for enhancing the readability of the manuscript. In the following, we address all the comments with a point-by-point reply. Correspondingly, we modify the manuscript based on these suggestions.

REFEREE: 1) The author should state in the main text what is the THz source, its power output, and the size of the focal spot. It is not clear to the reader.

AUTHORS’ REPLY: 1) We thank the reviewer for this comment. The terahertz frequency was tuned up to 0.30 THz (WR 2.8 Tripler) output from an IMPATT 100 GHz Diode, and 0.12 THz (WR 9 Tripler) from VDI multiplier connected to a 40-GHz microwave source. The 0.3 THz output beam was collimated by a set of two Polymethylpentene (TPX) lenses with 100mm effective focal length (EFL) resulting in a 5 mm diameter focal beam spot while its amplitude was modulated as a square wave at 1 kHz. The power output, calibrated by a TK100 power-meter, was 100 μW. THz-wave beam parameters are measured via NiTe₂-based device in combination with the scanning movement of a two-dimensional (horizontal and vertical directions) precision mechanical platform. The measurement of the THz beam spatial profile in the focal plane was also carried out by using a THz camera (see Figure R1). Following the referee’s suggestion, we have added a discussion regarding this issue in the *Methods* section of the main text and the revised supplementary information.

Figure R1. THz beam intensity profile along the vertical axis measured by the NiTe₂-based device. The inset shows a THz image of the focused beam by using a THz camera.

REFEREE: 2) The bulk of the supplemental section 7 should be discussed in the main text. It is also not clear how the current anisotropy comes from the calculation.

AUTHORS' REPLY: 2) We thank the referee for this suggestion. Following referee's comment, in the revised manuscript, we have moved part of the discussion previously reported in Sec. S7 of the supplementary information into the main text. We have also clarified the discussion regarding the origin of the current anisotropy in the revised manuscript (page10, line 197 to 207). Definitely, the anisotropy comes from the intrinsic C_{3v} rotation symmetry analysis¹⁻³, interplayed by the trigonal crystal field scattering and anisotropic effective mass of carriers, as recently measured in transport experiments on NiTe₂ and other materials of the same class (see also the Raman analysis in Fig. 1c). Besides, in type-II Dirac semimetals, the Fermi pocket enclosing the Dirac point is anisotropic, owing to the tilted Dirac node. Accordingly, the anisotropic scattering rate survives along different directions, corresponding to dissimilar current response. In calculations, the anisotropy comes from the integral of local current response, following the second-order skew scattering effect across the channel.

1 F. Fei, X. Bo, R. Wang, B. Wu, J. Jiang, D. Fu, M. Gao, H. Zheng, Y. Chen, X. Wang, H. Bu, F. Song, X. Wan, B. Wang, G. Wang. *Nontrivial Berry phase and type-II Dirac transport in the layered material PdTe₂*. *Phys. Rev. B*. 96, 041201(R) (2017).

2 Xu, C., Li, B., Jiao, W., Zhou, W., Qian, B., Sankar, R., Zhigadlo, N. D., Qi, Y., Qian, D., Chou, F.-C. & Xu, X. *Topological Type-II Dirac Fermions Approaching the Fermi Level in a Transition Metal Dichalcogenide NiTe₂*. *Chemistry of Materials* 30, 4823-4830, (2018).

3 Mukherjee, S., Jung, S. W., Weber, S. F., Xu, C., Qian, D., Xu, X., Biswas, P. K., Kim, T. K., Chapon, L. C., Watson, M. D., Neaton, J. B. & Cacho, C. *Fermi-crossing Type-II Dirac fermions and topological surface states in NiTe₂*. *Sci Rep* 10, 12957, (2020).

REFEREE: 3) Why the photocurrent is observed at THz frequencies is not clearly explained. It would be better if the authors can add a discussion on the frequency dependence.

AUTHORS' REPLY: 3) We thank the reviewer for having raised this issue. We point out that skew scattering under trigonal crystal-field, induced by the combining the inversion symmetry broken surface states and the chiral Bloch electrons, produces second-order rectification current.

Whenever the device is irradiated by electromagnetic radiation, the chiral Bloch electrons are driven back and forth by the ac oscillation of an electric field near the scattering site of trigonal crystal-field. Due to the inversion-symmetry breaking of topological surface states (TSSs), the scattered self-rotating wave-packet produces a directional photocurrent, whose sign changes with the polarization of oscillation electric field. This phenomenon is related to the spin-momentum locked TSSs. As a matter of fact, when self-rotating wave-packets are scattered by trigonal crystal-field, they will shift towards the same direction even with different spin-rotational directions, leading to the alignment of excess flow for photocurrent transport. In this regard, this phenomenon can be observed when the incident frequency is lower than the skew-scattering rate (or if they are close), so that the stationary directional photocurrent can be formed in the process after momentum relaxation. According to our results, the photoresponse decays at higher frequency. Specifically, THz frequencies are able to satisfy the skew-scattering in the scale of picoseconds^{1,2}. Correspondingly, we have modified the main text by adding lines 265-276 at Page 13-14.

¹ Das Sarma, S., Hwang, E. H. & Min, H. Carrier screening, transport, and relaxation in three-dimensional Dirac semimetals. *Physical Review B* 91, 035201, (2015).

² Lu, W., Ling, J., Xiu, F. & Sun, D. Terahertz probe of photoexcited carrier dynamics in the Dirac semimetal Cd_3As_2 . *Physical Review B* 98, (2018).

REFEREE: 4)

4. The sensitivity of the material can probably promise near-field THz photocurrent measurement. THz s-SNOM imaging can directly probe the proposed orientation dependent photocurrent, especially at the vicinity of the electrodes. Maybe the authors can comment on this?

AUTHORS' REPLY: 4) Thank you very much for bringing us such an interesting discussion. To our knowledge, SNOM is a very sophisticated tool that has been utilized formerly to study the near-field distribution from metallic plasmons, phonons, metamaterials or many others. Recent works have also validated that the near-field technique can be improved to scan photocurrent (mapping) locally across the channel

of a device within tens of nanometer spatial resolution, which is beneficial to understand the carrier dynamics under action of near-field^{1,2}.

Thus, we agree that near-field THz photocurrent microscopy is able to provide information on the photocurrent distribution, especially near the vicinity of contacts. Through scanning the tip of s-SNOM, the near-field from tip will interact strongly with chiral Bloch electrons. This interaction may be strongest near the material-metal interface, and it is probable that the largest photocurrent can be measured near electrodes^{3,4}. These experiments may lead to more intuitive imaging about the nonequilibrium process, open up alternative route to further studies from perspectives of the deep-subwavelength photonics. We briefly mentioned this issue in the revised version of the manuscript on page 7 line 138 to141.

1 Giordano, M. C., Viti, L., Mitrofanov, O. & Vitiello, M. S. Phase-sensitive terahertz imaging using room-temperature near-field nanodetectors. *Optica* 5, (2018).

2 Lundeberg, M. B., Gao, Y., Woessner, A., Tan, C., Alonso-Gonzalez, P., Watanabe, K., Taniguchi, T., Hone, J., Hillenbrand, R. & Koppens, F. H. Thermoelectric detection and imaging of propagating graphene plasmons. *Nat Mater* 16, 204-207, (2017).

3 Alonso-Gonzalez, P., Nikitin, A. Y., Gao, Y., Woessner, A., Lundeberg, M. B., Principi, A., Forcellini, N., Yan, W., Velez, S., Huber, A. J., Watanabe, K., Taniguchi, T., Casanova, F., Hueso, L. E., Polini, M., Hone, J., Koppens, F. H. & Hillenbrand, R. Acoustic terahertz graphene plasmons revealed by photocurrent nanoscopy. *Nat Nanotechnol* 12, 31-35, (2017).

4 Pogna, E. A. A., Asgari, M., Zannier, V., Sorba, L., Viti, L. & Vitiello, M. S. Unveiling the detection dynamics of semiconductor nanowire photodetectors by terahertz near-field nanoscopy. *Light: Science & Applications* 9, (2020).

REFEREE: 5) Many sentences can benefit from modifications/improvements in language, for example:

Line 91 “Our NiTe₂-based devices display a remarkably high sensitivity even at frequencies higher than those limited by the transit-time, with significant involvement of topologically protected surface and bulk bands.” It is not clear what does the authors mean by “significant involvement”

Line 313 “In Weyl semimetals, the rectification current is caused by the singularities of the Berry curvature and it is accompanied with poor power-linearity, whereas in Dirac semimetals there is not Berry curvature.” ♦ There is no Berry curvature.

AUTHORS’ REPLY: 5) We sincerely thank the reviewer for carefully reading the manuscript. Following referee’s suggestion, we have carefully revised the style in the manuscript, with the help of European coauthors of this manuscript.

————— REPLY TO REVIEWER 2 —————

REFEREE: "The manuscript considers second-order response in a topological semimetal NiTe₂ for a high-frequency rectifier. The authors performed materials characterization both experimentally and theoretically and fabricated a rectifier device

to show rectification and imaging in the THz frequency range. The use of a strongly-tilted type-II Dirac semimetal provides an interesting direction to this field. Overall, I think that the manuscript contains intriguing results supported by sound analyses. I would recommend for publication with some questions and comments below adequately answered."

AUTHORS' REPLY: We sincerely thank the reviewer for having appreciated our work. We also thank the reviewer for the constructive comments and suggestions, which have helped us to further improve the quality of the manuscript. All referee's comments and suggestions were carefully addressed at the best of our possibilities. Correspondingly, we revised the text of the manuscript in order to incorporate modifications to address referee's comments.

REFEREE: 1. The authors find the bulk Dirac point near the Fermi energy and state that it makes "a big challenge to signify the contribution of relativistic quasiparticles at low photon energies." As they use the topological surface state for rectification, the importance of the Dirac point close to the Fermi surface is not evident. They could mention the reason why it has to be.

AUTHORS' REPLY: 1) We thank the reviewer for raising this important point. There are two possible mechanisms that can give rise to the rectification current. Both of them rely on two different aspects of the NiTe₂ electronic structure:

- 1) **The presence of the type-II Dirac cone in nearness of the Fermi energy**
Bulk Dirac fermions are involved in the non-equilibrium carriers, due to the strong electromagnetic fields in the device geometry (see Fig. 2a of the manuscript). It can give rise to the rectification current.
- 2) **The presence of topological surface states near the Fermi energy**
Topological surface states in NiTe₂ arise from a band inversion in the conduction region. They can give rise to the rectification current, by means of the chiral Bloch scattering, as discussed in the manuscript.

While we have ruled out several other possibilities for the rectification current, we believe that a combination of both (1) and (2) is responsible for the observation of rectification current in our experiments.

To highlight the importance of the type-II Dirac fermions in NiTe₂, we note that in group X Pd- and Pt- based dichalcogenides, the bulk Dirac node lies deep below the Fermi level (~ 0.6 , ~ 0.8 and ~ 1.2 eV in PdTe₂, PtTe₂, and PtSe₂, respectively)¹⁻³, hindering their successful exploitations in technological applications related to photons with wavelength in the range of microwaves, Terahertz (THz) and far- and middle-infrared (FIR/MIR). In contrast, NiTe₂ hosts type-II Dirac fermions in nearness of the Fermi energy, as evident from Fig.1f of the main text. Our spin-resolved ARPES measurements explicitly demonstrate the existence of a pair of type-II Dirac nodes in NiTe₂ along the C₃ rotation axis, lying in the nearness of the Fermi energy.

This establishes NiTe₂ as a prime candidate for exploration of Dirac fermiology, with further possible applications (beyond those in our manuscript) in spintronic devices and ultrafast optoelectronics.

In Figure R2, we provide a comparison of band structures of MTe₂ (with M=Ni, Pd, Pt).

Figure R2: The band structure of NiTe₂, PdTe₂, and PtTe₂, showing the two Dirac points tilted in opposite directions, located on the A-Γ-A' axis. The dispersion around each of the Dirac points is isotropic in the horizontal S-D-T plane (parallel to the experimental Γ-K-M plane) and anisotropic and “tilted” along the Γ – A direction. A magnification of the band structure around one of the Dirac points is shown in panels (a), (f) and (i) for the isotropic S-D-T direction [marked by the red circle], and for anisotropic and “tilted” the Γ – A' direction in (b), (e) and (h) [marked by the green circle]. We have added this part of discussion to the revised supplementary information, also by adding to Supplementary Section S9 (page10,11) and Figure S10 (page 21).

- 1 Zhang, K., Yan, M., Zhang, H., Huang, H., Arita, M., Sun, Z., Duan, W., Wu, Y. & Zhou, S. *Experimental evidence for type-II Dirac semimetal in PtSe₂*. *Physical Review B* 96, 125102, (2017).
- 2 Clark, O. J., Neat, M. J., Okawa, K., Bawden, L., Marković, I., Mazzola, F., Feng, J., Sunko, V., Riley, J. M., Meevasana, W., Fujii, J., Vobornik, I., Kim, T. K., Hoesch, M., Sasagawa, T., Wahl, P., Bahramy, M. S. & King, P. D. C. *Fermiology and Superconductivity of Topological Surface States in PdTe₂*. *Physical Review Letters* 120, (2018).

3 Yan, M., Huang, H., Zhang, K., Wang, E., Yao, W., Deng, K., Wan, G., Zhang, H., Arita, M., Yang, H., Sun, Z., Yao, H., Wu, Y., Fan, S., Duan, W. & Zhou, S. Lorentz-violating type-II Dirac fermions in transition metal dichalcogenide PtTe₂. *Nat Commun* 8, 257, (2017).

REFEREE: 2. In line 154, they wrote, "The definite features of the nonlinear effects leading to the rectification are voltage-independent photocurrent at all incident frequencies". The meaning of this statement is unclear. They mentioned that applying a bias voltage does have some effects via the third-order process.

AUTHORS' REPLY: 2) We thank the reviewer for bringing this point to our attention. To avoid confusion, we have reworded this sentence "*The definite features of the nonlinear rectification are the voltage-independent photocurrent at all incident frequencies studied here.*" as "*The definite features of the second-order nonlinear rectification studied here are the self-driven photocurrent at all incident frequencies.*" The revision has been made from page 8 line 158 to 159.

By applying a bias voltage of above 10mV traversing across the channel in the low-temperature experiment, we find that the trend of temperature-dependent photocurrent is different from the that of zero-bias mode (self-driven mode), and it can be decomposed as the bias-dependent third-order process superimposed on the zero-bias response. The process of third-order process is that the nonequilibrium carriers under electrical bias are accelerated unilaterally from one side to another side of channel, due to the bias-induced asymmetry, resulting in a linear growth of photocurrent. In addition, it could be understood that the electric field E_{DC} plays the main role in tilting the Fermi levels, which results in the differently allowed momentum spaces for nonequilibrium carriers generation from opposite Dirac nodes in k-space, so that the non-equilibrium states cannot cancel out and, hence, will contribute to the net third-order photocurrent, Materials with low carrier tunable by E_{DC} . To address the referee's comment, we added lines 222-234 at page 11 of the revised manuscript.

REFEREE: 3)

3. Topological surface states are expected to appear both top and bottom surfaces. Presumably, only one surface contributes, or the contribution of one surface dominates. The authors should comment on the two-surface contributions.

AUTHORS' REPLY: 3) We thank the reviewer for this question. As the reviewer suggest, while the surface states are on both the surfaces, in our experiments the contribution of the top surface dominates. From the perspective of device fabrication process, the contacts are better on the top surface, and the sample is thicker than 100 nm, therefore the top surface contributes to most of the photocurrent, due to the static screening of potential at metal-material interface¹, as caused by the high density of Dirac fermions. Definitely, the joint contribution of topological surface states (TSSs)

from both top and bottom surfaces is feasible only for ultrathin thicknesses (~ 1 nm), especially in the presence of strong magnetic fields at very low temperature, for which the quantized orbit can imply the interplay of TSSs from both surfaces²⁻⁵.

Following the referee's suggestion, we have now explicitly mentioned this issue in the revised manuscript (page 7, line 144-146).

1 Xia, F., Perebeinos, V., Lin, Y. M., Wu, Y. & Avouris, P. *The origins and limits of metal-graphene junction resistance. Nat Nanotechnol* 6, 179-184, (2011).

2 Zhang, C., Narayan, A., Lu, S., Zhang, J., Zhang, H., Ni, Z., Yuan, X., Liu, Y., Park, J. H., Zhang, E., Wang, W., Liu, S., Cheng, L., Pi, L., Sheng, Z., Sanvito, S. & Xiu, F. *Evolution of Weyl orbit and quantum Hall effect in Dirac semimetal Cd₃As₂. Nat Commun* 8, 1272, (2017).

3 Zhang, C., Zhang, E., Wang, W., Liu, Y., Chen, Z. G., Lu, S., Liang, S., Cao, J., Yuan, X., Tang, L., Li, Q., Zhou, C., Gu, T., Wu, Y., Zou, J. & Xiu, F. *Room-temperature chiral charge pumping in Dirac semimetals. Nat Commun* 8, 13741, (2017).

4 Lu, W., Ling, J., Xiu, F. & Sun, D. *Terahertz probe of photoexcited carrier dynamics in the Dirac semimetal Cd₃As₂. Physical Review B* 98, (2018).

5 Zhang, C., Zhang, Y., Yuan, X., Lu, S., Zhang, J., Narayan, A., Liu, Y., Zhang, H., Ni, Z., Liu, R., Choi, E. S., Suslov, A., Sanvito, S., Pi, L., Lu, H. Z., Potter, A. C. & Xiu, F. *Quantum Hall effect based on Weyl orbits in Cd₃As₂. Nature* 565, 331-336, (2019).

REFEREE: 4. They attributed the observed rectification to skew scattering. It may be the most likely mechanism, but with only their observations, we cannot exclude other possibilities such as side jump due to symmetry reasons. Any comments would be helpful for readers.

AUTHORS' REPLY: 4) We thank the referee for this excellent point, with high potential interest for readership. We agree with the reviewer. Actually, based on experimental observations, a contribution arising from the side jump process cannot be ruled out. However, we note that a finite side-jump contribution generally arises, due to the interplay of Berry curvature and scattering. Being NiTe₂ a Dirac semimetal, it does not display a finite Berry curvature in the bulk. Thus, from a theoretical viewpoint, the role of the side-jump phenomena contributing to the observed rectification current seems unlikely for this specific case. Nevertheless, an extended discussion has been added in the main text (page 15, line 306-310) and supplementary Section S8 (page 8-9) in order to address this valuable suggestion by the reviewer.

REFEREE: 5. I could not see the reason why the zero bias rectification current decreases at low temperatures. It seems that thermally-smearred Fermi distribution degrades the effect at high temperatures.

AUTHORS' REPLY: 5) We appreciate the reviewer's valuable comment. Actually, finite temperature could indeed affect the intraband second-order photo-response through thermally smeared Fermi distribution function and the change of scattering times. It is reasonable from theoretical point of view that the zero-bias rectification will grow for highly doped semiconductors or semimetals at lower temperature, because of larger momentum relaxation time¹⁻⁴ of carrier transport or higher photon absorption.

In the specific realistic case of our system, it can be inferred from temperature-dependent output characteristic that the resistance has been changed by less than 10%, indicating the minimum change of material's transport property at lower temperature. On the other hand, the rectification current depends on the ratio (τ_s/τ_a) between symmetric scattering-time τ_s and asymmetry scattering-time τ_a (e. g. skew-scattering). Accordingly, it is possible that asymmetric scattering time is reduced at higher temperature as related to the skew scattering, leading to the growth of rectification efficiency⁵. It means that, in our specific case, the change of τ_a dominates the second-order photocurrent.

Anyway, it is desirable that higher mobility could imply superior rectification efficiency. Materials with low carrier density represent promising candidates for such purposes. Thus, it seems that thermally smeared distribution will not degrade significantly the zero-bias rectification current. Following the referee's suggestion, we have inserted a discussion on this issue in the revised supplementary information (page 6-7).

1 E. Deyo, L.E.Golub, E.L. Ivchenko, B. Spivak. *Semiclassical theory of the photogalvanic effect in non-centrosymmetric systems* (2009).

2 Sinova, J., Valenzuela, S. O., Wunderlich, J., Back, C. H. & Jungwirth, T. *Spin Hall effects. Reviews of Modern Physics* 87, 1213-1260, (2015).

3 Koniakhin, S. V. *Ratchet effect in graphene with trigonal clusters. The European Physical Journal B* 87, (2014).

4 Ferreira, A., Rappoport, T. G., Casalilla, M. A. & Castro Neto, A. H. *Extrinsic spin Hall effect induced by resonant skew scattering in graphene. Phys Rev Lett* 112, 066601, (2014).

5 Isobe, H., Xu, S.-Y. & Fu, L. *High-frequency rectification via chiral Bloch electrons. Sci.e Adv.* 6, eaay2497 (2020).

REFEREE: 6. The schematic figure in Fig. 3d needs explanation. Also, Fig. 3g, h do not have a figure legend or detailed description, so that the meaning is unclear.

AUTHORS' REPLY: 6) We thank the reviewer for this valuable suggestion. This was an oversight on our part. Following the referee's suggestion, we have added relevant explanation for the schematic figure in Fig. 3d (at page 13) and figure legends for Fig.3g, h (at page 14).

REFEREE: 7. In line 314, the authors mentioned the poor power linearity in Weyl semimetals without details or citations. It should be explained.

AUTHORS' REPLY: The corresponding sentence have been revised and explained clearly.

It will be re-expressed as:” *Explicitly, in Weyl semimetals the rectification current is caused by the singularities of the Berry curvature and it is accompanied by low excitation power, whereas in Dirac semimetals Berry curvature is absent¹⁷.*” (page 17 line 352-355)

REVIEWERS' COMMENTS

Reviewer #1 (Remarks to the Author):

The authors have successfully addressed all my comments. Again, I can recommend its publication in Nature Communications.

Reviewer #2 (Remarks to the Author):

Overall, the response to the previous report is satisfactory. I would recommend the manuscript for publication in the end; however, I'd like to point out a couple of points.

Related to comment 1 in the previous report, a Dirac point itself does not have finite Berry curvature, so that the fact that the Dirac point lies near the Fermi level is not directly related to the observed rectifying effect. Its topological properties can contribute, but I wanted this point to be clarified.

As for comment 6 in the previous report, I could not see modifications in Figs. 3g and 3h. The authors should check if the corrections are made properly.

————— **REPLY TO REVIEWER 1** —————

REFEREE: The authors have successfully addressed all my comments. Again, I can recommend its publication in Nature Communications.

AUTHORS' REPLY: We once again thank the reviewer for their professional comments, which will greatly help us improve the scientific quality of the manuscript.

————— **REPLY TO REVIEWER 2** —————

REFEREE: Overall, the response to the previous report is satisfactory. I would recommend the manuscript for publication in the end; however, I'd like to point out a couple of points.

AUTHORS' REPLY: We appreciate the reviewer for the time and valuable suggestion during reviewing our manuscript. Two points are valuable and represent important guidance to improve our work.

Related to comment 1 in the previous report, a Dirac point itself does not have finite Berry curvature, so that the fact that the Dirac point lies near the Fermi level is not directly related to the observed rectifying effect. Its topological properties can contribute, but I wanted this point to be clarified.

comment 1: We are grateful to the reviewer for his/her valuable suggestions. We are in fully agreement with the reviewers that topological property rather than position of Dirac point contributes to the rectifying effect. We have already claimed this issue at page 11 the manuscript in order to avoid misunderstanding. Actually, the topological properties of Dirac point and topological surface states (TSSs) is toggled with each other, and the rectifying is an immediately result from the nontrivial topology that leads to the skew scattering as we discussed in the whole manuscript.

As for comment 6 in the previous report, I could not see modifications in Figs. 3g and 3h. The authors should check if the corrections are made properly.

comment 6: We are grateful to the reviewer for his/her careful reading effort. we have checked carefully and added the corresponding figure legends in figure 3g and 3h (at page 25).

We once again thank the editors and reviewers for their dedication.